# Self-Reported Long COVID and Its Impact on COVID-19-Related Worries and Behaviors After Lifting the COVID-19 Restrictions in China

**DOI:** 10.3390/healthcare13030262

**Published:** 2025-01-29

**Authors:** Ziying Yang, Yihan Tang, Lingyu Kong, Xu Wang, Jinghua Li, Yuantao Hao, Zhiwei Wang, Jing Gu

**Affiliations:** 1Department of Medical Statistics, School of Public Health, Sun Yat-Sen University, No.74, Zhongshan Second Road, Guangzhou 510080, China; yangzy65@mail2.sysu.edu.cn (Z.Y.); tangyh39@mail2.sysu.edu.cn (Y.T.); kongly8@mail2.sysu.edu.cn (L.K.); wangx766@mail2.sysu.edu.cn (X.W.); lijinghua3@mail.sysu.edu.cn (J.L.); haoyt@bjmu.edu.cn (Y.H.); 2Guangzhou Joint Research Center for Disease Surveillance, Early Warning, and Risk Assessment, Guangzhou 510080, China; 3Sun Yat-Sen Global Health Institute, School of Public Health and Institute of State Governance, Sun Yat-Sen University, Guangzhou 510080, China; 4Guangdong Key Laboratory of Health Informatics, Guangzhou 510080, China; 5Research Center of Health Informatics, Sun Yat-Sen University, Guangzhou 510080, China; 6Peking University Center for Public Health and Epidemic Preparedness and Response, Peking University, Beijing 100191, China; 7Department of 12320 Health Hotline, Guangzhou Center for Disease Control and Prevention, Guangzhou 510120, China

**Keywords:** long COVID, COVID-19-related worries, mental health, preventive measures

## Abstract

Objective: Since the lifting of the COVID-19 restrictions in China in November 2022, there has been a notable surge in the COVID-19 infection rate. Little is known about the prevalence of long COVID among the general adult population and its impact on COVID-19-related worries and behaviors after the policy change. Methods: This cross-sectional study recruited 1530 adults with prior COVID-19 infection in Guangzhou from February to March 2023. Logistic regression analyses and trend analyses were performed to investigate the associations between long COVID- and COVID-19-related worries and preventive behaviors. Results: The estimated prevalence of long COVID among adults in China was 18.0% (95% confidence interval: 16.1% to 20.0%). Common long COVID symptoms included cough (60.7%), fatigue (47.6%), dyspnea (34.5%), palpitation (26.2%), and insomnia (25.1%). Adjusted for background variables, individuals with long COVID exhibited higher level of COVID-19-related worries compared to those who had fully recovered from the infection (reference: without long COVID; adjusted odds ratios ranged from 1.87 to 2.55, all *p* values < 0.001). Participants primarily expressed worries regarding the potential for COVID-19 reinfection, the impact of the pandemic on daily life, the increasing number of COVID-19 cases and deaths, and the capacity of the healthcare system. While long COVID did not statistically significantly affect their preventive behaviors. Conclusions: Long COVID was prevalent among the general adult population in China after lifting the COVID-19 restrictions, and it had a significant impact on COVID-19-related worries. This study highlights the importance of monitoring the mental health of individuals with long COVID and developing targeted intervention strategies to improve their adherence to preventive measures.

## 1. Introduction

Since the first outbreak in January 2020, the COVID-19 infection has exhibited multiple waves across various countries worldwide, attributed to the ongoing evolution of the viruses [1,2,3]. As of 18 October 2023, there have been a total of 771.42 million reported infection cases and 6.97 million reported death cases worldwide [4]. The multiple waves of the COVID-19 pandemic have resulted in a multitude of adverse consequences globally, including healthcare system overload [5,6,7], heightened psychological burdens [8,9], and significant economic losses [10]. In addition to experiencing acute-phase symptoms, COVID-19 patients may also develop post-acute sequelae, commonly referred to as long COVID [11]. According to the US Centers for Disease Control and Prevention (CDC), long COVID is defined as signs, symptoms, and conditions that develop or persist for a minimum of four weeks following the acute phase of COVID-19 infection [12]. The World Health Organization (WHO) defines long COVID as the continuation or development of new symptoms three months after the initial SARS-CoV-2 infection, with these symptoms lasting for at least two months and with no other plausible explanation [13].

A meta-analysis has shown that the pooled prevalence of long COVID ranges from 39.0% to 46.0% (the US CDC criterion) [14]. Common long COVID symptoms include fatigue, memory problems, dyspnea, sleep disorders, and joint pain, among others [14]. Notably, the majority of long COVID patients experience a combination of multiple symptoms simultaneously [15]. Infection with different variants of SARS-CoV-2 may entail varying risk of developing long COVID and can result in distinct symptoms profiles if long COVID does occur [16,17]. According to another meta-analysis, the pooled prevalence of long COVID among individuals infected with the original strain, the alpha variant, the delta variant, and the omicron variant was recorded as 52.1%, 65.8%, 34.6%, and 28.4%, respectively (WHO criterion) [18].

After China eased the COVID-19 control measures in November 2022, there was a significant surge in the infection rate, which reached 92.3% by 31 January 2023 [19]. Preceding the relaxation of COVID-19 control measures, several studies have reported the prevalence of long COVID in the Chinese population [20,21,22,23], with disparities in the findings attributed to differences in dominant viral strains and population characteristics (e.g., age and the severity of underlying health conditions). Wong et al. surveyed 2717 COVID-19 patients who were infected with the original strain, alpha, beta, delta, and omicron variants during year 2020 to 2022 [20], and the occurrence rate of long COVID reached 90.4%. In Luo’s study of 6242 individuals who were infected with the omicron variants during 31 December 2021 to 6 May 2022, 55.0% of the respondents developed long COVID [22]. In a larger-scale study of 21,799 adults infected with the omicron variants during 24 March to 7 June 2022, the prevalence of long COVID was 8.9% [21]. Considering the widespread infection of nearly all residents in China after the relaxation of the COVID-19 response, it is essential to evaluate the prevalence of long COVID during this specific period.

In the context of major public health events, appropriate emotional responses play an essential role in maintaining not only individual well-being but also social stability [24,25]. Amidst the COVID-19 pandemic, worry has emerged as a common negative emotion [26]. Individuals primarily expressed worries about perceived risk of infection for oneself and one’s family [27], disruptions to daily life [28], potential COVID-19-related events in the future [29], challenges in obtaining medical resources [30], and the long-term impacts on physical health [31] at this period. Research has shown that individuals who have experienced more severe COVID-19 symptoms tend to exhibit elevated levels of worry [32]. Moreover, heightened worry has been associated with increased risk of various conditions, such as coronary heart disease and eating disorders [33,34]. Individuals who developed long COVID are more prone to heightened COVID-19-related concerns due to the extended duration of COVID-19 symptoms, in contrast to those who have fully recovered from the infection. However, there is currently a lack of studies specifically reporting on COVID-19-related worries within the long COVID group.

Existing studies have demonstrated the effectiveness of protective behaviors in reducing the risk of COVID-19 reinfection [35]. Long COVID patients may be at elevated risk of reinfection due to the weakened immunity [36,37,38]. A cross-sectional study has identified the association between the frequency of COVID-19 reinfection and the history of long COVID [36]. Furthermore, the currently predominant Omicron variant is characterized by high transmissibility and immune evasion [39], leading to a higher reinfection rate among individuals infected with Omicron compared to those infected with other variants [40,41]. Long COVID is pathological condition characterized by persistent immunosuppression [42]. In Fournelle et al.’s intra-host analysis in 2024, distinct mutants of the virus were discovered from an immunosuppressed patient, suggesting the possibility of the emergence of mutations under immunosuppression [43]. Thus, individuals with long COVID are suggested to proactively adopt protective behaviors to reduce the risk of reinfection and prevent the virus from spreading to others. Such protective measures include mask-wearing, hand washing, social distancing, and avoiding public transportation and social gatherings [44,45,46]. However, the adherence to preventive behaviors among the long COVID population is unknown.

This study aims to assess the prevalence of long COVID and the common symptoms in the Chinese population after relaxing the COVID-19 restrictions. It also investigates the impact of long COVID on COVID-19-related worries and preventive behaviors among individuals with prior infection. Additionally, the study assesses the influence of severity of long COVID symptoms on COVID-19-related worries and behaviors. This study seeks to provide empirical evidence to facilitate the management of the long COVID population and the development of tailored health interventions.

## 2. Materials and Methods

### 2.1. Study Design

This study employed a repeated cross-sectional survey design, targeting residents aged 18 years and above, who resided in Guangzhou, and self-reported having had infected with COVID-19. The online questionnaire survey was carried out through the Guangzhou Health Hotline 12320, a government-operated health service platform that focuses on health education, promotion, and the conduction of surveys [47]. The survey commenced in November 2022, coinciding with a rapid rise in COVID-19 infection rates in Guangzhou. The frequency of data collection was adjusted based on the severity of the epidemic: it was conducted weekly in November and December 2022, biweekly in January 2023, and then monthly thereafter. Questions concerning long COVID were incorporated into the survey during February and March 2023, and the data gathered over these two months were analyzed for this study.

### 2.2. Data Collection

The 12320 system randomly generated mobile phone numbers by considering the demographic distribution across the 11 districts of Guangzhou. In the two rounds, 40,000 and 45,000 phone numbers were selected, and invitations via SMS were sent to these selected numbers. Over the next three days, the survey link was distributed to the selected phone numbers to gather data via an electronic questionnaire. The questionnaire began with an overview of the study’s goals, details, sponsoring organizations, and the estimated time needed to complete it. Participants were assured that their involvement was entirely voluntary and that all personal information would be treated with the utmost confidentiality. Participants indicated their informed consent to participate in the survey by clicking to begin.

In both rounds of the survey, the click-through rate for SMS invitations was 12.5%, while the completion rate for questionnaires from those who accessed the link was 33.8%. The overall response rate, calculated as the number of completed questionnaires divided by the total number of invitations sent, was 2.3%, with rates of 2.2% and 2.3% for the first and second rounds, respectively. Of the collected 1931 questionnaires, 4 were less than 18 years old, and 397 were not infected with COVID-19; therefore, a total of 1530 samples were included in the analyses. Figure 1 shows a flow chart of included samples.

### 2.3. Measures

#### 2.3.1. Self-Reported Long COVID and Symptoms

Self-reported long COVID was assessed by asking the participants: “Are you currently experiencing long COVID, defined as the persistence of COVID-19 symptoms or the development of new symptoms one month after initial infection with COVID-19” (0 = No, 1 = Yes). For those who reported having long COVID, the presence of long COVID symptoms, including cough, fatigue, dyspnea, palpitation, insomnia, cognitive impairment, depression/anxiety, dizziness, chest pain, headache, joint pain, tinnitus/earache, nausea, diarrhea, and rash, was further inquired about. The number of long COVID symptoms for each participants was calculated and recorded into an ordered categorical variable with three levels (0, 1–2, and ≥3) according to the literature [48]. Symptoms with prevalence over 25% were defined as prevalent symptoms [49], and participants were classified into three group: with, without the prevalent long COVID symptoms, or without long COVID.

#### 2.3.2. COVID-19-Related Worries

COVID-19-related concerns were measured using five items drawn from the existing literature [27,28,29,30,31]. Sample items included, “Over the past week, how much have you worried about COVID-19 infection for yourself and your family?” and “Over the past week, how much have you been concerned about the pandemic’s effects on the daily lives of you and your family?” The other three items assessed the level of concern regarding potential surging of COVID-19 cases and deaths in Guangzhou and the limited healthcare system capacity. All five items were rated on a 5-point Likert scale (1 = not worried at all, 2 = not very worried, 3 = worried, 4 = somewhat worried, 5 = very worried). The responses were recorded into two groups: not worried (1 and 2) and worried (3, 4, and 5). A total score of COVID-19-related worries was calculated for each participant by adding up the scores of all five items. The total score ranges from 5 to 25, with a higher score indicating higher levels of concern. The Cronbach’s α was 0.924 in this study.

#### 2.3.3. COVID-19-Related Preventive Behaviors

Based on existing literature [44,45,46], participants’ engagement in COVID-19 preventive behaviors over the previous week was measured using five items. These included practices such as wearing masks in public, washing hands immediately upon returning home, maintaining a one-meter distance in line, avoiding public transportation, and avoiding social gatherings. These items were rated on a 4-point Likert scale (1 = not well-implemented [0–40% of time], 2 = partially implemented [41–70% of time], 3 = mostly implemented [71–90% of time], 4 = strictly implemented [91–100% of time]). The responses were recorded into two groups: not highly implemented (1 and 2) and highly implemented (3 and 4). A total score of COVID-19-related preventive behaviors was calculated for each participant by adding up the five items. The total score ranged from 5 to 20, with higher scores indicating more frequent implementation of the five behaviors. The Cronbach’s α was 0.762 in this study.

#### 2.3.4. Background Characteristics

Socio-demographic characteristics were collected, including sex, age, education level, occupation, marital status, income level, and the duration of living in Guangzhou. COVID-19-infection-related characteristics included the presence of chronic diseases, time of last COVID-19 infection, and the corresponding testing method. The survey time was also recorded.

### 2.4. Statistical Analysis

Descriptive statistics were calculated for participants’ background characteristics. Univariate association analyses between the presence of long COVID and background variables, as well as binary items of COVID-19-related worries and behaviors, were performed using a Chi-square test. Mann–Whitney U tests were conducted for univariate association analyses between the presence of long COVID and the total scores. To analyze the trend across the number of long COVID symptoms and the presence of prevalent symptoms, a Cochran–Armitage test for trend was conducted for binary items, while a Jonckheere’s trend test was used for the total scores. Phi correlation was used to examine the correlation between long COVID symptoms and COVID-19-related worries. Logistic regression models and linear regression models were used to test the associations between long COVID status and COVID-19-related worries, as well as preventive behaviors, adjusting for potential background confounders (with *p* < 0.1 in the univariate analyses). Adjusted odds ratio (AORs) and regression coefficient (β) were estimated. The respective 95% confidence intervals (CIs) were also reported. *p* < 0.05 was considered statistically significant. Data were analyzed using R Statistical Software, version 4.2.3 [50].

## 3. Results

### 3.1. Background Characteristics of Participants

Of the 1530 participants infected with COVID-19, more than half were male (55.5%), less than 40 years old (68.0%), had attained a college or above degree (73.9%), were currently employed (78.0%), were married (52.2%), had an income level >685 USD (58.2%), lived in Guangzhou for more than 6 months (95.2%), and had no chronic diseases (85.9%). In total, 14.5% of the participants were infected before December 2022, while 79.7% were infected during December 2022, and 5.8% were infected after December 2022. Over half of them were tested positive with a rapid antigen test (RAT) (52.2%), 15.4% with a nucleic acid amplification test (NAAT), 14.3% with both the RAT and NAAT, while 18.2% with a symptom-based assessment. In total, 45.9% of the participants were interviewed in February 2023, while 54.1% were interviewed in March 2023 (Table 1). Apart from age groups and time of last COVID-19 infection, there were no significant differences in other background characteristics between the two rounds of survey data (Appendix A).

### 3.2. Prevalence and Symptoms of Long COVID

The overall prevalence of long COVID in February and March 2023 was 18.0% (275/1530, 95% CI: 16.1% to 20.0%): 18.9% and 17.1% (*p* = 0.412) during the two rounds of surveys, respectively. The prevalence was slightly higher in females compared to males (20.3% vs. 16.1%, *p* = 0.043) and in individuals with higher education levels compared to those with lower education levels (19.6% vs. 13.3%, *p* = 0.006). Other socio-demographic characteristics, COVID-19-infection-related characteristics, and survey time did not differentiate the presence of long COVID (Table 1).

Among the 15 surveyed long COVID-19 symptoms, the prevalent symptoms were cough (60.7%, 167/275), fatigue (47.6%, 131/275), dyspnea (34.5%, 95/275), palpitation (26.2%, 72/275), and insomnia (25.1%, 69/275) (Figure 2 and Appendix A). Of the 275 participants with long COVID, 23.3% had only one symptom, 24.3% had two or three symptoms, while 52.4% had more than three symptoms. Individuals with at least one prevalent symptom (cough, fatigue, dyspnea, palpitation, or insomnia) accounted for 90.5% of all participants with long COVID.

### 3.3. COVID-19-Related Worries and Preventive Behaviors

A significant part of all participants (35.6% to 50.5%) reported currently being worried due to COVID-19: 50.5% were worried about a surge in COVID-19 cases in Guangzhou, followed by being worried about the healthcare system capacity (47.5%), the surge in COVID-19 deaths (47.0%), daily life being affected by the pandemic (45.3%), and reinfection (35.6%). As for preventive behaviors, over 70% of participants well implemented the behaviors of wearing masks in public (84.4%) and washing hands immediately upon returning home (78.3%), 53.9% could maintain a one-meter distance in line, and 37.1% and 35.6% could avoid public transportation and social gatherings, respectively. The median and IQR of the total score of COVID-19-related worries and preventive behaviors are listed in Table 2 and Table 3, respectively.

### 3.4. Associations Between Long COVID and Worries

In the univariate analyses, compared to those without long COVID, individuals with long COVID had a higher prevalence of all five worry items related to COVID-19 (all *p* values < 0.001, Table 2); the difference in the total score of COVID-19-related worries was also statistically significant between the two groups (*p* < 0.001). Regarding the number of long COVID symptoms (0, 1–2, and ≥3), for the presence of prevalent symptoms (no long COVID, no prevalent symptoms, with prevalent symptoms) and worries (items and total score), all the trend analyses were positive and statistically significant (all *p* values < 0.001, Table 2). The correlations analyses indicated that nine of the sixteen surveyed long COVID symptoms were positively correlated with all five items of COVID-19-related worries (Appendix A). These symptoms include cough, fatigue, dyspnea, palpitations, insomnia, cognitive impairment, depression/anxiety, and diarrhea, most of which are prevalent symptoms. In contrast, symptoms such as dizziness, chest pain, headache, tinnitus/earache, and nausea were only positively correlated with worries about reinfection and daily life affected by the pandemic.

After adjusting for potential background confounders, all associations between the presence of long COVID and worries (items and total score) remained significant (reference: without long COVID; AORs ranged from 1.87 to 2.55, *p* < 0.001, Table 4). Participants with three or more long COVID symptoms (reference: without long COVID; AORs ranged from 2.40 to 3.91, *p* < 0.001) and with prevalent long COVID symptoms (reference: without long COVID; AORs ranged from 1.90 to 2.69, *p* < 0.001) were more likely to be worried (Table 4).

### 3.5. Associations Between Long COVID and Preventive Behaviors

Individuals with and without long COVID did not differ in five items and total score of preventive behaviors (Table 3). Except for the behavior of wearing masks in public (*p* < 0.05, Table 3), all other items and the total score presented no significant trend across the number of long COVID symptoms and presence of prevalent symptoms. After adjusting for background variables, the associations between preventive behaviors (items and total score) and long COVID remained non-significant (Table 5).

## 4. Discussion

This study presented the first report on the prevalence of long COVID among individuals with prior infection in China after lifting the COVID-19 restrictions (18.0%). The consistent estimates across the two rounds of the surveys suggested relative stability of the long COVID prevalence. Common long COVID symptoms included cough, fatigue, dyspnea, palpitation, and insomnia. COVID-19-related worries were prevalent, particularly worries about surging COVID-19 cases in Guangzhou and about the healthcare system capacity. Long COVID, the number of long COVID symptoms, and prevalent symptoms were positively associated with COVID-19-related worries. Nevertheless, the implementation of preventive behaviors was relatively low, except for mask-wearing and hand washing. All considered preventive behaviors showed no significant association with long COVID, the number of long COVID symptoms, or prevalent symptoms.

All individuals in the study were infected with COVID-19 after October 2022. According to the WHO’s surveillance report, the dominant strains during this period were the Omicron variants B.A.5.2 and B.F.7 [51]. The estimated prevalence of long COVID in this study was lower than the 28.4% of Omicron variants reported in a meta-analysis [18]. This discrepancy could be attributed to a decrease in pathogenicity with the ongoing evolution of the virus and the differences in the surveyed samples [52]. Luo et al. reported an estimated long COVID prevalence of 55.0% among the Chinese population also during the epidemic of Omicron variants [22]. But the proportions of individuals aged over 60 (26.8%) and with chronic diseases (32.4%) were higher than the corresponding proportions of 5.0% and 14.1% in the present study. The relatively high proportion of young individuals in the present study indicated the under-representation of the elderly population. The extrapolation of the long COVID prevalence to the entire population could potentially lead to an underestimation. Considering the surge in the COVID-19 infection rate to as high as 92.3% in China following the COVID-19 policy change, there is potentially a substantial long COVID population at present. Thus, it is crucial to assess the health status of the long COVID population. Health guidance, including the use of low-dose aripiprazole to alleviate brain fog and probiotics to address gastrointestinal symptoms [53], should be provided to them in a timely manner.

This study found that the long COVID population exhibits a high degree of COVID-19-related worries, exceeding the proportion of being very or extremely worried about COVID-19 during the early outbreak (19.8%), as estimated in a UK study [54]. Additionally, it was discovered that long COVID patients demonstrated elevated levels of worries compared to those who have fully recovered from COVID-19. Specifically, individuals experiencing prevalent long COVID symptoms are likely to exhibit heightened levels of worries across various aspects. The findings indicate that worries about reinfection and the impact of the pandemic on daily life are two significant factors correlated with long COVID symptoms. As evidenced in previous studies, bearing a heavy symptom burden, long COVID patients tend to show heightened levels of worry [55]. This psychological state of worry is linked to numerous adverse health outcomes. For instance, a cohort study involving 1759 men found that heightened levels of worry correlated with a substantially increased risk of coronary heart disease (CHD) [33]. The study reported multivariate adjusted relative risk of 2.41 for nonfatal myocardial infarction and 1.48 for total CHD. Therefore, it is crucial to continuously monitor both the physical and mental health of the long COVID population, particularly those experiencing prevalent symptoms. To address the psychological stress faced by this group, it is necessary to promote intervention programs, such as internet-based self-help interventions and worry-reduction ecological momentary interventions, both of which have been shown to be effective in practice [56,57]. Such programs should be made widely accessible to support the mental well-being of individuals affected by long COVID.

This study discovered that, after easing the COVID-19 control measures, individuals previously infected with COVID-19 exhibited relatively high rates of implementing wearing masks (84.4%) and handwashing (78.3%), while adherence to other preventive behaviors was comparatively low. The implementation of the five studied preventive behaviors has declined compared to the early phase of the pandemic when adherence rates for all preventive behaviors were over 90.0% [58]. This finding is consistent with previous research [59]. The present study also revealed that long COVID patients and individuals who have fully recovered from COVID-19 exhibited no difference in implementing these preventive behaviors. However, the COVID-19 pandemic remains widespread [60], with continual emergence of new variants, such as XBB.1.5 and EG.5 [61,62]. Considering the high reinfection risk and the possibility of spreading the virus to others, it is necessary for long COVID patients to actively adopt protective behaviors.

The present study has several limitations. First, the response rate in this study was low. Though similar results have been reported in other studies, where the response rates using SMS invitation were less than 2% [63,64], it was not possible to obtain the characteristics from the randomly selected individuals who did not participate in the study, potentially resulting in selection bias in the survey data. Given that the collected sample were younger relative to the general adult population, there may be a social desirability bias in the participants’ responses. Second, the study sample exhibited higher proportions of individuals aged below 40 (68.0%) and those with a bachelor’s degree or higher (73.9%), and the proportion of individuals aged over 60 is relatively low. As a result, the findings of this study might not be fully generalizable to the general adult population. Third, the identification of long COVID was based on self-report from participants, possibly resulting in an overestimation of the prevalence. Fourth, the association analyses between long COVID and COVID-19-related worries, as well as preventive behaviors only incorporated socio-demographic variables, three variables related to COVID-19 infection, and survey time, neglecting other potential confounders, such as the individual’s COVID-19 vaccination status, the severity of acute infection, and parental status [11,20]. Fifth, the total scores of COVID-19-related worries and behaviors in this study were self-constructed according to the existing literature, which are not comparable to scores from other studies. Lastly, the cross-sectional design of this study restricts its ability to infer causal relationships.

## 5. Conclusions

After the relaxation of the COVID-19 restrictions, there has been a relatively high proportion of long COVID patients in China. Long COVID heightened the worries about the pandemic but did not affect the adherence to preventive measures. Individuals with long COVID may bear significant psychological burdens, further underscoring the need for monitoring their mental health. The persistent symptoms associated with long COVID, coupled with uncertainties about how to manage their condition, can lead to heightened levels of anxiety, depression, and feelings of isolation. These factors can significantly impact their overall psychological well-being. Acknowledging and addressing these psychological implications is essential for delivering comprehensive care and support to this population. Future studies need to investigate the causal factors for psychology-related symptoms among long COVID patients. Furthermore, there is a need for tailored health promotion campaigns targeting long COVID population, emphasizing their susceptibility and the importance of adopting preventive behaviors. Important directions for future research include the investigation of the mechanisms behind adherence to protective measures, as well as the potential development of phobias or obsessive symptomatology related to COVID-19.

## Figures and Tables

**Figure 1 healthcare-13-00262-f001:**
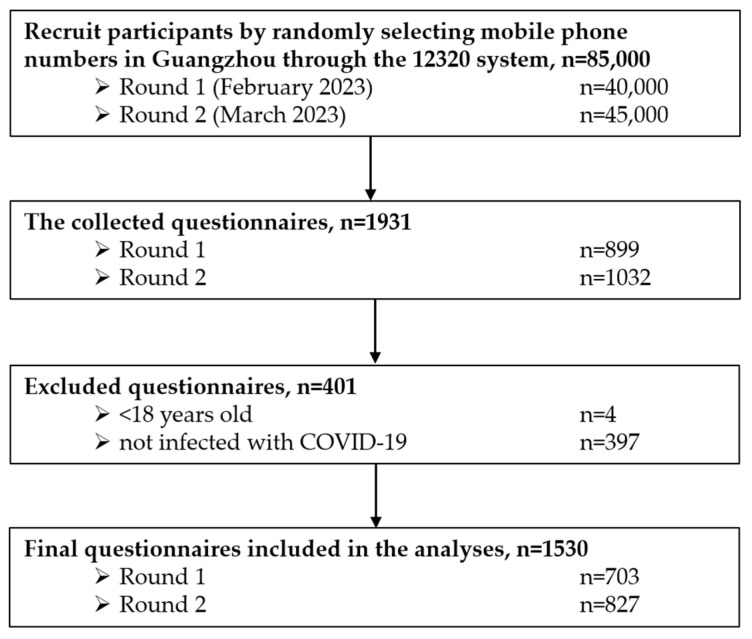
Flowchart of included sample.

**Figure 2 healthcare-13-00262-f002:**
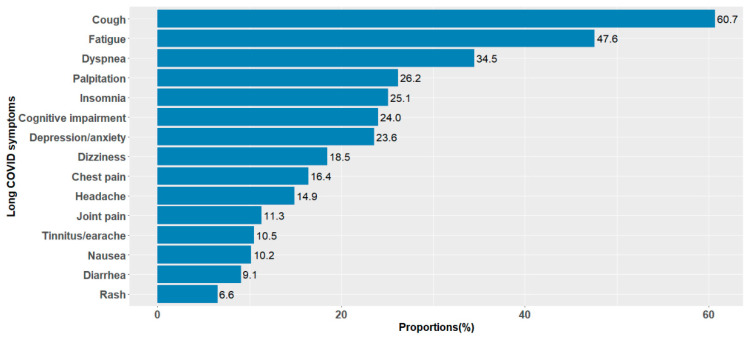
Distribution of long COVID symptoms among participants who self-reported long COVID (*n* = 275).

**Table 1 healthcare-13-00262-t001:** Background characteristics of participants by long COVID status.

	All (*n* = 1530)	Long COVID, (*n*, %)		*p* Value
		No (*n* = 1255)	Yes (*n* = 275)	
Socio-demographic characteristics				
Sex				0.043
Male	849 (55.5)	712 (83.9)	137 (16.1)	
Female	681 (44.5)	543 (79.7)	138 (20.3)	
Age groups (years)				0.312
18–30	599 (39.2)	498 (83.1)	101 (16.9)	
31–40	441 (28.8)	350 (79.4)	91 (20.6)	
41–50	269 (17.6)	218 (81.0)	51 (19.0)	
51–60	145 (9.4)	125 (86.2)	20 (13.8)	
>60	76 (5.0)	64 (84.2)	12 (15.8)	
Education level				0.006
Below college	399 (26.1)	346 (86.7)	53 (13.3)	
College or above	1131 (73.9)	909 (80.4)	222 (19.6)	
Occupation				0.081
Business/Service staff	374 (24.4)	312 (83.4)	62 (16.6)	
Administrative officials	217 (14.2)	185 (85.3)	32 (14.7)	
Technical practitioners	392 (25.6)	302 (77.0)	90 (23.0)	
Production/Transportation staff or others	211 (13.8)	177 (83.9)	34 (16.1)	
Retired/Unemployed	202 (13.2)	170 (84.2)	32 (15.8)	
Students	134 (8.76)	109 (81.3)	25 (18.7)	
Marital status				0.619
Unmarried	655 (42.8)	540 (82.4)	115 (17.6)	
Married	798 (52.2)	655 (82.1)	143 (17.9)	
Divorced/Widowed or others	77 (5.0)	60 (77.9)	17 (22.1)	
Income level (USD)				0.084
No income	213 (13.9)	183 (85.9)	30 (14.1)	
<685	426 (27.8)	355 (83.3)	71 (16.7)	
685–1371	522 (34.1)	429 (82.2)	93 (17.8)	
>1371	369 (24.1)	288 (78.0)	81 (22.0)	
Duration living in Guangzhou				0.385
>6 months	1456 (95.2)	1191 (81.8)	265 (18.2)	
≤6 months	74 (4.8)	64 (86.5)	10 (13.5)	
COVID-19 infection-related characteristics				
Had chronic diseases				0.278
No	1314 (85.9)	1084 (82.5)	230 (17.5)	
Yes	216 (14.1)	171 (79.2)	45 (20.8)	
Time of last COVID-19 infection				0.901
December 2022	1220 (79.7)	998 (81.8)	222 (18.2)	
October 2022–November 2022	221 (14.5)	183 (82.8)	38 (17.2)	
After December 2022	89 (5.8)	74 (83.1)	15 (16.9)	
Testing method of COVID-19				0.529
RAT ^a^	798 (52.2)	651 (81.6)	147 (18.4)	
NAAT ^b^	235 (15.4)	197 (83.8)	38 (16.2)	
Both RAT and NAAT	219 (14.3)	174 (79.5)	45 (20.5)	
Symptom-based assessment	278 (18.2)	233 (83.8)	45 (16.2)	
Survey time				0.412
February 2023	703 (45.9)	570 (81.1)	133 (18.9)	
March 2023	827 (54.1)	685 (82.8)	142 (17.2)	

^a^: RAT: rapid antigen test; ^b^: NAAT: nucleic acid amplification test.

**Table 2 healthcare-13-00262-t002:** COVID-19-related worries by long COVID status.

	Worry About Reinfection		Worry about Daily Life Affected by the Pandemic		Worry About Surge in COVID-19 Cases in Guangzhou		Worry About Surge in COVID-19 Deaths in Guangzhou		Worry About Healthcare System Capacity		Total Scores of COVID-19-Related Worries	
	**Row (%)**	$\chi^{2}$	**Row (%)**	$\chi^{2}$	**Row (%)**	$\chi^{2}$	**Row (%)**	$\chi^{2}$	**Row (%)**	$\chi^{2}$	**Median (IQR)**	**MWU**
All (*n*, %)	545 (35.6)		693 (45.3)		773 (50.5)		719 (47.0)		727 (47.5)		12 [9,10,11,12,13,14,15,16]	
Long COVID		**41.9 *****		**42.9 *****		**22.4 *****		**18.5 *****		**24.1 *****		**125,934 *****
No	31.9		41.4		47.6		44.4		44.5		11 [8,9,10,11,12,13,14,15,16]	
Yes	52.7		63.3		63.6		58.9		61.1		15 [10,11,12,13,14,15,16,17,18,19]	
Number of long COVID symptoms		**−7.36 ***^a^**		**−7.28 ***^a^**		**−5.12 ***^a^**		**−5.03 ***^a^**		**−5.92 ***^a^**		**231,348 ***^b^**
0	31.9		41.4		47.6		44.4		44.5		11 [8,9,10,11,12,13,14,15,16]	
1 or 2	42.0		53.4		58.0		50.4		49.6		13 [10,11,12,13,14,15,16,17]	
≥3	62.5		72.2		68.8		66.7		71.5		16 [12,13,14,15,16,17,18,19,20]	
Presence of prevalent symptoms		**−6.64 ***^a^**		**−6.76 ***^a^**		**−4.70 ***^a^**		**−4.54 ***^a^**		**−5.18 ***^a^**		**222,958 ***^b^**
No long COVID	31.9		41.4		47.6		44.4		44.5		11 [8,9,10,11,12,13,14,15,16]	
No prevalent symptoms	42.3		50.0		65.4		46.2		46.2		12.5 [10–17.5]	
With prevalent symptoms	53.8		64.7		63.5		60.2		62.7		15 [11,12,13,14,15,16,17,18,19]	

$\chi^{2}$, Chi-square test. IQR, inter-quartile range. MWU, Mann–Whitney U test statistic. ^a^, Cochran–Armitage test statistic for trend. ^b^, Jonckheere’s trend test statistic. ***, *p* < 0.001. Test statistics with *p* < 0.05 were in bold. Prevalent symptoms included cough, fatigue, dyspnea, palpitation, and insomnia. Non-prevalent symptoms included cognitive impairment, depression/anxiety, dizziness, chest pain, headache, joint pain, tinnitus/earache, nausea, diarrhea, and rash.

**Table 3 healthcare-13-00262-t003:** Preventive behaviors by long COVID status.

	Wear Masks in Public		Wash Hands Immediately Upon Returning Home		Maintain a One-Meter Distance in Line		Avoid Public Transportations		Avoid Social Gatherings		Total Scores of Preventive Behaviors	
	Row (%)	$\chi^{2}$	Row (%)	$\chi^{2}$	Row (%)	$\chi^{2}$	Row (%)	$\chi^{2}$	Row (%)	$\chi^{2}$	Median (IQR)	MWU
All (*n*, %)	1292 (84.4)		1198 (78.3)		824 (53.9)		568 (37.1)		544 (35.6)		13 [11,12,13,14,15,16]	
Long COVID		3.6		1.3		1.0		0.2		1.5 × 10 ^−3^		169,010
No	83.6		77.7		54.5		36.8		35.6		13 [11,12,13,14,15]	
Yes	88.4		81.1		50.9		38.5		35.3		13 [11,12,13,14,15,16]	
Number of long COVID symptoms		**−2.06 *^a^**		−0.89 ^a^		0.55 ^a^		−0.63 ^a^		−0.35 ^a^		186,260 ^b^
0	83.6		77.7		54.5		36.8		35.6		13 [11,12,13,14,15]	
1 or 2	87.0		83.2		46.6		37.4		31.3		13 [11,12,13,14,15,16]	
≥3	89.6		79.2		54.9		39.6		38.9		13 [11,12,13,14,15,16]	
Presence of prevalent symptoms		**−2.14 *^a^**		−1.23 ^a^		1.05 ^a^		−0.60 ^a^		0.02 ^a^		179,702 ^b^
No long COVID	83.6		77.7		54.5		36.8		35.6		13 [11,12,13,14,15]	
No prevalent symptoms	80.8		80.8		50.0		34.6		30.8		12.5 [10.2–15.8]	
With prevalent symptoms	89.2		81.1		51.0		39.0		35.7		13 [11,12,13,14,15,16]	

$\chi^{2}$, Chi-square test. IQR, inter-quartile range. MWU, Mann–Whitney U test statistic. ^a^, Cochran–Armitage test statistic for trend. ^b^, Jonckheere’s trend test statistic. *, *p* < 0.05. Test statistics with *p* < 0.05 were in bold. Prevalent symptoms include cough, fatigue, dyspnea, palpitation, and insomnia. Non-prevalent symptoms included cognitive impairment, depression/anxiety, dizziness, chest pain, headache, joint pain, tinnitus/earache, nausea, diarrhea, and rash.

**Table 4 healthcare-13-00262-t004:** Adjusted associations between long COVID status and COVID-19-related worries ^†^.

	Worry About Reinfection	Worry About Daily Life Affected by the Pandemic	Worry About Surging COVID-19 Cases in Guangzhou	Worry About Surging COVID-19 Deaths in Guangzhou	Worry About Healthcare System Capacity	Total Scores of COVID-19 Related Worries
	AOR (95%CI) ^a^	AOR (95%CI) ^b^	AOR (95%CI) ^c^	AOR (95%CI) ^d^	AOR (95%CI) ^e^	β (95%CI) ^f^
Self-reported long COVID						
No	1.00	1.00	1.00	1.00	1.00	1.00
Yes	**2.47 (1.89, 3.24) *****	**2.55 (1.93, 3.36) *****	**1.95 (1.48, 2.58) *****	**1.87 (1.42, 2.46) *****	**1.94 (1.48, 2.55) *****	**2.52 (1.81, 3.22) *****
Number of long COVID symptoms						
0	1.00	1.00	1.00	1.00	1.00	1.00
1 or 2	**1.64 (1.13, 2.37) ****	**1.67 (1.15, 2.42) ****	**1.58 (1.09, 2.30) ***	1.36 (0.94, 1.97)	1.27 (0.88, 1.83)	**1.28 (0.32, 2.25) ****
≥3	**3.63 (2.52, 5.21) *****	**3.91 (2.65, 5.78) *****	**2.40 (1.64, 3.51) *****	**2.55 (1.75, 3.72) *****	**3.00 (2.04, 4.42) *****	**3.65 (2.72, 4.57) *****
Presence of prevalent symptoms						
No long COVID	1.00	1.00	1.00	1.00	1.00	1.00
No prevalent symptoms	1.77 (0.80, 3.93)	1.57 (0.71, 3.46)	**2.50 (1.09, 5.75) ***	1.26 (0.57, 2.79)	1.27 (0.58, 2.81)	1.78 (−0.31, 3.87)
With prevalent symptoms	**2.56 (1.93, 3.39) *****	**2.69 (2.01, 3.60) *****	**1.90 (1.42, 2.54) *****	**1.95 (1.47, 2.61) *****	**2.04 (1.53, 2.71) *****	**2.59 (1.86, 3.33) *****

^†^, COVID-19-related worries as the outcome. ^a^, Adjusted for education level, income level, presence of chronic diseases, and survey time. ^b^, Adjusted for education level, occupation, marital status, income level, presence of chronic diseases, and survey time. ^c^, Adjusted for sex, age, occupation, income level, and survey time. ^d^, Adjusted for sex, age, occupation, income level, and survey time. ^e^, Adjusted for sex, age, income level, and survey time. ^f^, Adjusted for sex, age, income level, duration living in Guangzhou, presence of chronic diseases, and survey time. AOR, Odds ratios adjusting for significant background variables (*p* < 0.1). Univariate *p* values between background variables and COVID-19-related worries were shown in Appendix A. *, *p* < 0.05. **, *p* < 0.01. ***, *p* < 0.001. AOR, β and 95%CI with *p* < 0.05 were in bold. Prevalent symptoms include cough, fatigue, dyspnea, palpitation, and insomnia. Non-prevalent symptoms included cognitive impairment, depression/anxiety, dizziness, chest pain, headache, joint pain, tinnitus/earache, nausea, diarrhea, and rash.

**Table 5 healthcare-13-00262-t005:** Adjusted associations between long COVID status and preventive behaviors ^†^.

	Wear Masks in Public	Wash Hands Immediately upon Returning Home	Maintain a One-Meter Distance in Line	Avoid Public Transport	Avoid Social Gatherings	Total Scores of Preventive Behaviors
	AOR (95%CI) ^a^	AOR (95%CI) ^b^	AOR (95%CI) ^c^	AOR (95%CI) ^d^	AOR (95%CI) ^e^	β (95%CI) ^f^
Self-reported long COVID						
No	1.00	1.00	1.00	1.00	1.00	1.00
Yes	1.45 (0.96, 2.19)	1.17 (0.82, 1.66)	0.87 (0.66, 1.15)	1.10 (0.83, 1.46)	1.01 (0.76, 1.34)	0.17 (−0.28, 0.62)
Number of persistent symptoms						
0	1.00	1.00	1.00	1.00	1.00	1.00
1 or 2	1.30 (0.75, 2.26)	1.37 (0.83, 2.27)	0.72 (0.49, 1.05)	1.02 (0.69, 1.51)	0.86 (0.58, 1.29)	0.01 (−0.61, 0.64)
≥3	1.62 (0.91, 2.86)	1.01 (0.64, 1.59)	1.03 (0.72, 1.48)	1.18 (0.81, 1.70)	1.15 (0.79, 1.66)	0.31 (−0.29, 0.91)
Presence of prevalent symptoms						
No long COVID	1.00	1.00	1.00	1.00	1.00	1.00
No prevalent symptoms	0.93 (0.34, 2.60)	1.13 (0.40, 3.17)	0.84 (0.37, 1.91)	0.80 (0.35, 1.86)	0.81 (0.34, 1.94)	−0.48 (−1.83, 0.86)
With prevalent symptoms	1.55 (1.00, 2.44)	1.17 (0.81, 1.69)	0.87 (0.66, 1.16)	1.14 (0.85, 1.53)	1.03 (0.77, 1.38)	0.24 (−0.23, 0.71)

**^†^**, Preventive behaviors as the outcome. ^a^, Adjusted for sex, age, occupation, marital status, duration living in Guangzhou, and survey time. ^b^, Adjusted for sex, age, occupation, marital status, income level, duration living in Guangzhou, presence of chronic diseases, time when tested positive, and survey time. ^c^, Adjusted for age, education level, occupation, marital status, income level, presence of chronic diseases, and survey time. ^d^, Adjusted for sex, age, education level, occupation, marital status, income level, and positive testing method. ^e^, Adjusted for age, education level, occupation, marital status, income level, presence of chronic diseases, positive testing method, and survey time. ^f^, Adjusted for sex, age, education level, occupation, marital status, income level, presence of chronic diseases, positive testing method, and survey time. AOR, Odds ratios adjusting for significant background variables (*p* < 0.1). Univariate *p* values between background variables and preventive behaviors were shown in Appendix A. Prevalent symptoms include cough, fatigue, dyspnea, palpitation, and insomnia. Non-prevalent symptoms included cognitive impairment, depression/anxiety, dizziness, chest pain, headache, joint pain, tinnitus/earache, nausea, diarrhea, and rash.

## Data Availability

The data presented in this study are available on request from the corresponding author (the data are not publicly available due to privacy or ethical restrictions).

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
