# Peer review of "Self-Reported Long COVID and Its Impact on COVID-19-Related Worries and Behaviors After Lifting the COVID-19 Restrictions in China"

_healthcare, 2025, doi:10.3390/healthcare13030262_

Round 1
Reviewer 1 Report
Comments and Suggestions for Authors
The paper addresses a significant issue regarding the impact of long-standing COVID on the Chinese population after the end of pandemic restrictions and how this has affected COVID-19 concerns and preventive behaviour. The study is cross-sectional and was conducted on a sample of 1530 adult subjects who responded to and filled out an online questionnaire. The sample size allowed to obtain interesting and statistically significant results that were generalisable to a broader population. Overall, I find the work interesting and believe it may be considered for publication in your journal after making minor changes, as indicated below.
ABSTRACT
Lines 31-32: "individuals with long COVID exhibited higher levels of COVID-19-related worries". Since the text reports the prevalent symptoms associated with long-COVID, it would be helpful to specify the main types of concerns that emerge from the sample analysed. Authors could use the introduction (lines 84 - 88) to describe the prevailing problems.
MATERIALS AND METHODS
The description of the experimental design and data collection procedures is well-structured and detailed. The measurements made are precise and supported by existing literature. I suggest including a flow chart to illustrate data collection process (timing) and the study structure (number of subjects recruited, number of subjects who responded, number of subjects excluded). This would improve the clarity of the section in general, making it easier to understand even for readers who are not very familiar with the subject.
RESULTS
The results are clearly shown; however, special attention should be given to the arrangement of tables 2, 3, 4, and 5 to present the reported data more easily.
Only prevalent symptoms have been reported in the table legends. Supplementing these with non-prevalent symptoms may give a better overview of the situation.
DISCUSSION
Lines 342-345: A tangible clinical example may be helpful in supporting the authors' statements.
CONCLUSION
While the conclusions match the results derived, they could be expanded. I recommend discussing more the psychological implications of long-term COVID concerns, especially for mental health and psychological well-being.
Author Response
Comments 1: The paper addresses a significant issue regarding the impact of long-standing COVID on the Chinese population after the end of pandemic restrictions and how this has affected COVID-19 concerns and preventive behaviour. The study is cross-sectional and was conducted on a sample of 1530 adult subjects who responded to and filled out an online questionnaire. The sample size allowed to obtain interesting and statistically significant results that were generalisable to a broader population. Overall, I find the work interesting and believe it may be considered for publication in your journal after making minor changes, as indicated below.
Response 1: Thanks for the comments.
Comments 2: ABSTRACT: Lines 31-32: "individuals with long COVID exhibited higher levels of COVID-19-related worries". Since the text reports the prevalent symptoms associated with long-COVID, it would be helpful to specify the main types of concerns that emerge from the sample analysed. Authors could use the introduction (lines 84 - 88) to describe the prevailing problems.
Response 2: Thank you for the valuable suggestions. We agree that it is essential to clearly outline the main types of worries that emerge from the analyzed sample in the Abstract. Accordingly, we have added a sentence specifying the main worries (Lines 34-36 on Page 1), shown as follows:
“Participants primarily expressed worries regarding the potential for COVID-19 reinfection, the impact of the pandemic on daily life, the increasing number of COVID-19 cases and deaths, and the capacity of the healthcare system.”
Comments 3: MATERIALS AND METHODS: The description of the experimental design and data collection procedures is well-structured and detailed. The measurements made are precise and supported by existing literature. I suggest including a flow chart to illustrate data collection process (timing) and the study structure (number of subjects recruited, number of subjects who responded, number of subjects excluded). This would improve the clarity of the section in general, making it easier to understand even for readers who are not very familiar with the subject.
Response 3: Thank you for the advice. We agree that including a flowchart will enhance readers’ comprehension of our data collecting process and the structure of the study. As a result, we have incorporated the flowchart as Figure 1 in the revised manuscript. Additionally, we have revised the relevant sentence in the Data collection subsection (Line 152 on Page 4), shown as follows:
“Figure 1 shows a flow chart of included samples.”
Comments 4: RESULTS: The results are clearly shown; however, special attention should be given to the arrangement of tables 2, 3, 4, and 5 to present the reported data more easily. Only prevalent symptoms have been reported in the table legends. Supplementing these with non-prevalent symptoms may give a better overview of the situation.
Response 4: Thanks for the comments. We have reviewed the arrangements of tables 2-5 to ensure they align with the order of the reported results in the manuscript. In accordance with your suggestions, we have also expanded the table legends for tables 2-5 to include descriptions of non-prevalent symptoms (Lines 288-289 on Page 9, Lines 294-295 on Page 9, and Lines 306-307 on Page 10), shown as follows:
“Non-prevalent symptoms included cognitive impairment, depression/anxiety, dizziness, chest pain, headache, joint pain, tinnitus/earache, nausea, diarrhea, and rash.”
Comments 5: DISCUSSION: Lines 342-345: A tangible clinical example may be helpful in supporting the authors' statements.
Response 5: Thanks for the insightful suggestions. We agree that providing a tangible example would enhance the support for our assertion regarding the association between the psychological state of worry and an increased risk of adverse health outcomes. Accordingly, we have revised this section as follows (Lines 369-373 on Page 12):
“This psychological state of worry is linked to numerous adverse health outcomes. For instance, a cohort study involving 1,759 men found that heightened levels of worry correlated with a substantially increased risk of coronary heart disease (CHD)[56]. The study reported multivariate adjusted relative risk of 2.41 for nonfatal myocardial infarction and 1.48 for total CHD.”
[56] Kubzansky, L.D., et al., Is worrying bad for your heart? A prospective study of worry and coronary heart disease in the Normative Aging Study. Circulation, 1997. 95(4): 818-824.
Comments 6: CONCLUSION: While the conclusions match the results derived, they could be expanded. I recommend discussing more the psychological implications of long-term COVID concerns, especially for mental health and psychological well-being.
Response 6: Thanks for the advices. As you suggested, we have revised the conclusion section to expand upon the psychological implications of long COVID, as follows (Lines 415-421 on Page 13):
“Individuals with long COVID may bear significant psychological burdens, further underscoring the need for monitoring their mental health. The persistent symptoms associated with long COVID, coupled with uncertainties about how to manage their condition, can lead to heightened levels of anxiety, depression, and feelings of isolation. These factors can significantly impact their overall psychological well-being. Acknowledging and addressing these psychological implications is essential for delivering comprehensive care and support to this population.”
Reviewer 2 Report
Comments and Suggestions for Authors
The paper focused on self-reported long COVID19 worries and symptoms in Chinese population.
I found the paper interesting, with an high numerosity, even if I note some minor issues to review.
Procedure
Is there an Ethical committe approval?
In the socio-demographic information I think that it could be important to understand also if the participants had or not children because the literature stressed that having children could add stresses for people increasing the health worries related to Covid symptomatology.
Was a difference along age also found in the worries perceptions or in long Covid symptoms?
I suggest also to run some correlations between symptoms and worries because probably these constructs are associated and could influence one to each other.
What about possible psychological and physical interventions to dampen worries and symptoms? In the discussion it could be added these comments.
Future studies have also to take into consideration in the worries also anxiety and depression symptoms as the literature stressed. Probably it is important to assess possible development of phobias or obsessive symptomatology related to Covid19.
Another limit to be added in the discussion is the social desirability effect that could influence the responses of the participants, also because they were principally young.
Author Response
Comments 1: The paper focused on self-reported long COVID19 worries and symptoms in Chinese population. I found the paper interesting, with an high numerosity, even if I note some minor issues to review.
Response 1: Thanks for the comments.
Comments 2: Procedure: Is there an Ethical committee approval?
Response 2: Yes, our study received approval from the Institutional Review Board of the School of Public Health, Sun Yat-sen University on February 4, 2020, with the approval code No. 2020-005. This information is detailed in the Institutional Review Board Statement subsection (Lines 435-437 on Page 13).
Comments 3: In the socio-demographic information I think that it could be important to understand also if the participants had or not children because the literature stressed that having children could add stresses for people increasing the health worries related to Covid symptomatology.
Response 3: Thanks for the comments. We acknowledged that having children may contribute to increased psychological stress and heightened health-related worries, particularly in the context of COVID-19. However, the main goal of our study was to investigate the impact of long COVID specifically on COVID-19 related worries. Consequently, parental status was not within the scope of our investigation, and we did not collect this information. We have acknowledged this as a limitation in the Discussion section (Line 408 on Page 13).
Comments 4: Was a difference along age also found in the worries perceptions or in long Covid symptoms?
Response 4: Thanks for your comments. In response to your inquiry, we analyzed the distribution of COVID-19 related worries and long COVID symptoms across various age groups (shown in Table 1 and Table 2 below). The results indicated that there were no statistically significant differences in most COVID-19 related worries, nor in any of the reported long COVID symptoms, across these age groups. Although there was a statistically difference in concerns about surging COVID-19 cases in Guangzhou and healthcare system, we did not observe a linear trend. As a result, we have chosen not to include a detailed discussion of this finding in the revised manuscript.
Table 1 Distribution of COVID-19 related worries across different age groups.
|
COVID-19 related worries |
18-30 |
31-40 |
41-50 |
51-60 |
>60 |
pa |
|
Worry about reinfection (%) |
35.9 |
33.8 |
36.8 |
33.8 |
43.4 |
0.548 |
|
Worry about daily life affected by the pandemic (%) |
42.9 |
44.4 |
50.2 |
46.9 |
48.7 |
0.329 |
|
Worry about surging COVID-19 cases in Guangzhou (%) |
53.1 |
50.1 |
46.8 |
42.1 |
61.8 |
0.025 |
|
Worry about surging COVID-19 deaths in Guangzhou (%) |
49.9 |
47.2 |
43.9 |
37.9 |
51.3 |
0.076 |
|
Worry about healthcare system capacity (%) |
49.9 |
50.6 |
43.1 |
36.6 |
47.4 |
0.017 |
a, Chi-square test
Table 2 Distribution of long COVID symptoms across different age groups (%).
|
Long COVID symptoms |
18-30 |
31-40 |
41-50 |
51-60 |
>60 |
pa |
|
Cough |
10.7 |
12.9 |
11.2 |
8.3 |
5.3 |
0.247 |
|
Fatigue |
8.5 |
10.2 |
8.9 |
3.5 |
7.9 |
0.168 |
|
Dyspnea |
7.0 |
6.6 |
6.7 |
1.4 |
5.3 |
0.147 |
|
Insomnia |
4.7 |
6.1 |
3.4 |
2.8 |
5.3 |
0.359 |
|
Cognitive impairment |
4.2 |
5.0 |
3.7 |
5.5 |
5.3 |
0.874 |
|
Depression/anxiety |
3.7 |
5.2 |
6.0 |
2.1 |
2.6 |
0.240 |
|
Dizziness |
5.3 |
4.3 |
3.0 |
2.8 |
2.6 |
0.390 |
|
Chest pain |
3.8 |
3.2 |
2.2 |
3.5 |
4.0 |
0.805 |
|
Headache |
3.0 |
3.4 |
2.6 |
1.4 |
4.0 |
0.744 |
|
Joint pain |
3.8 |
2.5 |
0.7 |
2.1 |
2.6 |
0.125 |
|
Tinnitus/earache |
1.8 |
1.6 |
1.9 |
3.5 |
4.0 |
0.483 |
|
Nausea |
1.8 |
2.3 |
1.9 |
0.7 |
2.6 |
0.791 |
|
Diarrhea |
2.3 |
2.5 |
0.4 |
1.4 |
0 |
0.152 |
|
Rash |
2.2 |
2.3 |
0.7 |
0 |
0 |
0.127 |
a, Chi-square test
Comments 5: I suggest also to run some correlations between symptoms and worries because probably these constructs are associated and could influence one to each other.
Response 5: Thanks for the valuable advices. As you suggested, we conducted analyses to examine the correlation between long COVID symptoms and COVID-19 related worries, as detailed in Table 3 below. The results revealed that nine of the sixteen surveyed long COVID symptoms were positively correlated with all five items of COVID-19-related worries. These symptoms include cough, fatigue, dyspnea, palpitations, insomnia, cognitive impairment, depression/anxiety and diarrhea, most of which are prevalent. In contrast, symptoms such as dizziness, chest pain, headache, tinnitus/earache, and nausea were only positively correlated with worries about reinfection and daily life affected by the pandemic. Based on these findings, it is suggested that individuals experiencing prevalent long COVID symptoms are likely to exhibit heightened levels of worries across various aspects. Among the five items of COVID-19 related worries assessed, worries about reinfection and worries about daily life affected by the pandemic are two significant aspects correlated with long COVID symptoms.
We have included this additional content into the Materials and Methods section (Lines 209-210 on Page 5), the Results section (Lines 269-276 on Page 8), and the Discussion section (Lines 364-368 and Lines 374-375 on Page 12) of the revised manuscript. Table 3 has been added as Table S3 in the supplementary material. Thank you again for your valuable suggestions.
Table 3 Correlations between long COVID symptoms and COVID-19 related worries (.
|
|
Worry about reinfection |
Worry about daily life affected by the pandemic |
Worry about surging COVID-19 cases in Guangzhou |
Worry about surging COVID-19 deaths in Guangzhou |
Worry about healthcare system capacity |
|
Cough |
15.03*** |
15.81*** |
3.51* |
5.73** |
7.88*** |
|
Fatigue |
21.18*** |
25.70*** |
12.91*** |
15.90*** |
21.37*** |
|
Dyspnea |
24.47*** |
25.71*** |
13.57*** |
12.81*** |
14.71*** |
|
Palpitations |
14.76*** |
12.30*** |
6.56** |
6.13** |
6.74** |
|
Insomnia |
15.49*** |
16.41*** |
5.38** |
13.01*** |
8.61*** |
|
Cognitive impairment |
23.85*** |
23.27*** |
10.81*** |
14.20*** |
10.81*** |
|
Depression/anxiety |
24.96*** |
24.53*** |
11.66*** |
12.12*** |
16.20*** |
|
Dizziness |
5.04** |
5.79** |
0.40 |
1.09 |
2.81* |
|
Chest pain |
2.73* |
3.06* |
0.85 |
2.42 |
3.04* |
|
Headache |
4.45** |
4.16* |
0.94 |
1.80 |
4.22* |
|
Joint pain |
2.79* |
1.04 |
0.06 |
0.74 |
1.96 |
|
Tinnitus/earache |
6.69** |
5.03** |
1.36 |
4.36** |
3.01* |
|
Nausea |
5.87** |
4.46** |
1.07 |
3.84* |
1.68 |
|
Diarrhea |
8.56*** |
5.52** |
3.66* |
6.41** |
4.67** |
|
Rash |
0.70 |
0.82 |
0.85 |
0.61 |
2.30 |
*, p<0.05. **, p<0.01. ***, p<0.001.
Comments 6: What about possible psychological and physical interventions to dampen worries and symptoms? In the discussion it could be added these comments.
Response 6: Thanks for the advices. We agree that discussing possible psychological and physical interventions to dampen worries and symptoms is necessary. We have revised the Discussion section as follows (Lines 357-359 and Lines 375-380 on Page 12):
“Health guidance, including the use of low-dose aripiprazole to alleviate brain fog and probiotics to address gastrointestinal symptoms[53], should be provided to them in a timely manner.”
“To address the psychological stress faced by this group, it is necessary to promote intervention programs, such as Internet-Based Self-Help Interventions and worry-reduction ecological momentary interventions, both of which have been shown to be effective in practice[57, 58]. Such programs should be made widely accessible to support the mental well-being of individuals affected by long COVID.”
[53] Davis, H.E., et al., Long COVID: major findings, mechanisms and recommendations. Nature Reviews Microbiology, 2023. 21(3): 133-146.
[57] Heckendorf, H., D. Lehr, and L. Boß, Effectiveness of an internet-based self-help intervention versus public mental health advice to reduce worry during the COVID-19 pandemic: a pragmatic, parallel-group, randomized controlled trial. Psychotherapy and Psychosomatics, 2022. 91(6): 398-410.
[58] Versluis, A., et al., Effectiveness of a smartphone-based worry-reduction training for stress reduction: A randomized-controlled trial. Psychology & health, 2018. 33(9): 1079-1099.
Comments 7: Future studies have also to take into consideration in the worries also anxiety and depression symptoms as the literature stressed. Probably it is important to assess possible development of phobias or obsessive symptomatology related to Covid19.
Response 7: Thanks for your advices. We agree that it is important to assess possible development of phobias or obsessive symptomatology related to COVID-19. We have revised the Discussion section as follows (Lines 426-428 on Page 13):
“Important directions for future research include the investigation of the mechanisms behind adherence to protective measures, as well as the potential development of phobias or obsessive symptomatology related to COVID-19.”
Comments 8: Another limit to be added in the discussion is the social desirability effect that could influence the responses of the participants, also because they were principally young.
Response 8: Thank you for the comment. We agree that social desirability bias may be a potential limitation, particularly given that our sample primarily comprises younger individuals. We have added the following limitation in the Discussion section (Lines 397-399 on Page 12):
“Given that the collected sample were younger relative to the general adult population, there may be a social desirability bias in the participants’ responses.”

Reviewer 3 Report
Comments and Suggestions for Authors
Intro: Good definition and description of long Covid and symptoms. Description of Covid-19 related worries and note that increased worries related to more severe Covid-19 symptoms and to the development of long Covid. Long Covid seems to predispose to another Covid infection; would expect these individuals to adhere to preventive behaviors. Purposes of the study – prevalence of long Covid after relaxing Covid-19 restrictions, impact of long Covid and severity of symptoms on Covid related worries and preventive behaviors.
Materials + Methods: Repeated cross-sectional survey with an electronic questionnaire. Socio-demographic information along with presence of other diseases obtained. Questions pertained to presence of long Covid symptoms, Covid-19 related worries and the implementation of Covid-related preventive behaviors. Questions, scoring and grouping all well described. Methods of statistical analyses described.
Results: Table I clearly shows demographics and Covid-19 infection date and symptoms. It would be helpful to have a Table to represent the information presenting the prevalence of symptoms. Fig. 1 belongs up here. There should be a Table for prevalence of symptoms relating to sociodemographic factors. Associations of Covid-19 with worries and preventive behaviors for all participants was well presented. Associations between long Covid and worries and long Covid and preventive behaviors were well presented in Tables and accompanying text.
Discussion: Many of the participants were worried about Covid; those with long Covid were more worried than those without. However, preventive behaviors (except for mask wearing and hand washing) were not well instituted in any group and there were no significant differences between those with and without long Covid. These findings are consistent with findings of other studies. Noted limitations-low response rate, lower age and higher level of education of respondents, self-reporting of symptoms, no information re: vaccine status, cross-sectional study, other potential confounders not evaluated.
Conclusions: After the relaxation of the COVID-19 restrictions, there has been a relatively high
proportion of long COVID patients in China. Long COVID heightened the worries but did not affect the adherence to preventive measures. Programs to monitor mental health of long Covid patients and to improve instituting preventive behavior are needed.
Author Response
Comments 1: Intro: Good definition and description of long Covid and symptoms. Description of Covid-19 related worries and note that increased worries related to more severe Covid-19 symptoms and to the development of long Covid. Long Covid seems to predispose to another Covid infection; would expect these individuals to adhere to preventive behaviors. Purposes of the study – prevalence of long Covid after relaxing Covid-19 restrictions, impact of long Covid and severity of symptoms on Covid related worries and preventive behaviors.
Response 1: Thanks for the comments.
Comments 2: Materials + Methods: Repeated cross-sectional survey with an electronic questionnaire. Socio-demographic information along with presence of other diseases obtained. Questions pertained to presence of long Covid symptoms, Covid-19 related worries and the implementation of Covid-related preventive behaviors. Questions, scoring and grouping all well described. Methods of statistical analyses described.
Response 2: Thanks for the comments.
Comments 3: Results: Table I clearly shows demographics and Covid-19 infection date and symptoms. It would be helpful to have a Table to represent the information presenting the prevalence of symptoms. Fig. 1 belongs up here. There should be a Table for prevalence of symptoms relating to sociodemographic factors. Associations of Covid-19 with worries and preventive behaviors for all participants was well presented. Associations between long Covid and worries and long Covid and preventive behaviors were well presented in Tables and accompanying text.
Response 3: Thanks for the comments. As you suggested, we have included a table that presents the prevalence of long COVID symptoms, along with their distribution across various sociodemographic factors. This table can be found as Table S2 in the supplementary material. We appreciate your insights, which have helped enhance the clarity of our findings.
Comments 4: Discussion: Many of the participants were worried about Covid; those with long Covid were more worried than those without. However, preventive behaviors (except for mask wearing and hand washing) were not well instituted in any group and there were no significant differences between those with and without long Covid. These findings are consistent with findings of other studies. Noted limitations-low response rate, lower age and higher level of education of respondents, self-reporting of symptoms, no information re: vaccine status, cross-sectional study, other potential confounders not evaluated.
Response 4: Thanks for the comments.
Comments 5: Conclusions: After the relaxation of the COVID-19 restrictions, there has been a relatively high proportion of long COVID patients in China. Long COVID heightened the worries but did not affect the adherence to preventive measures. Programs to monitor mental health of long Covid patients and to improve instituting preventive behavior are needed.
Response 5: Thanks for the comments.
